# BACH: grand challenge on breast cancer histology images

**Guilherme Aresta**[*1,2]                                      GUILHERME.ARESTA@INESCTEC.PT
**Teresa Araújo**[*1,2]                                              TFARAUJO@INESCTEC.PT
**Aurélio Campilho**[2]                                               CAMPILHO@FE.UP.PT
**Catarina Eloy**[3]                                                   CELOY@IPATIMUP.PT
**António Polónia**[3]                                              APOLONIA@IPATIMUP.PT
**Paulo Aguiar**[3]                                                PAULOAGUIAR@INEB.UP.PT

[1] *INESC TEC - Institute for Systems and Computer Engineering, Technology and Science, Portugal*

[2] *Faculty of Engineering of University of Porto, Portugal*

[3] *Instituto de Investigação e Inovação em Saúde (i3S), Universidade do Porto, Portugal*

**Editors:** Under Review for MIDL 2019

## Abstract

The Grand Challenge on BreAst Cancer Histology images (BACH) aimed at the classification and localization of clinically relevant histopathological classes in microscopy and whole-slide images from a large annotated dataset, specifically compiled and made publicly available for the challenge. A total of 64 submissions, out of 677 registrations, effectively entered the competition. The submitted algorithms improved the state-of-the-art in automatic classification of breast cancer with microscopy images to an accuracy of 87%, with convolutional neural networks being the most successful methodology. Detailed analysis of the results allowed the identification of remaining challenges in the field and recommendations for future developments. The BACH dataset remains publicly available to promote further improvements to the field of automatic classification in digital pathology. [1]

**Keywords:** Breast cancer, Histology, Digital pathology, Challenge, Deep learning

## 1. Challenge description

The Grand Challenge on BreAst Cancer Histology images (BACH) was organized with the $15^{th}$ Int. Conf. on Image Analysis and Recognition (ICIAR) to promote advances in the automatic classification of H&E histopathological breast biopsy images. BACH had two parts (P-**A** and **B**) aimed at the identification of four classes: 1) Normal, 2) Benign, 3) *in situ* and 4) Invasive carcinoma. P-**A**'s goal was the image-level classification of micropscopy images. For that, 400 train and 100 test samples (2.0×1.5 kpixel, 2 experts' image-wise annotations), with equal class distribution, were provided. For comparison, 3 external experts were also asked to label this test set. P-**B** aimed at the pixel-level classification of whole-slide (WSI) images and 10 annotated and 20 non-annotated train and 10 test samples ([40 62]×[28 45] kpixel, 1 expert) were provided. Prior to the test set release, participants submitted a paper to ICIAR reporting their approach and expected performance. The test set labels are kept hidden and BACH is still online [2].

---

\* Contributed equally

1. This paper summarizes a homonymous work currently under review for *Medical Image Analysis*

2. https://iciar2018-challenge.grand-challenge.org/

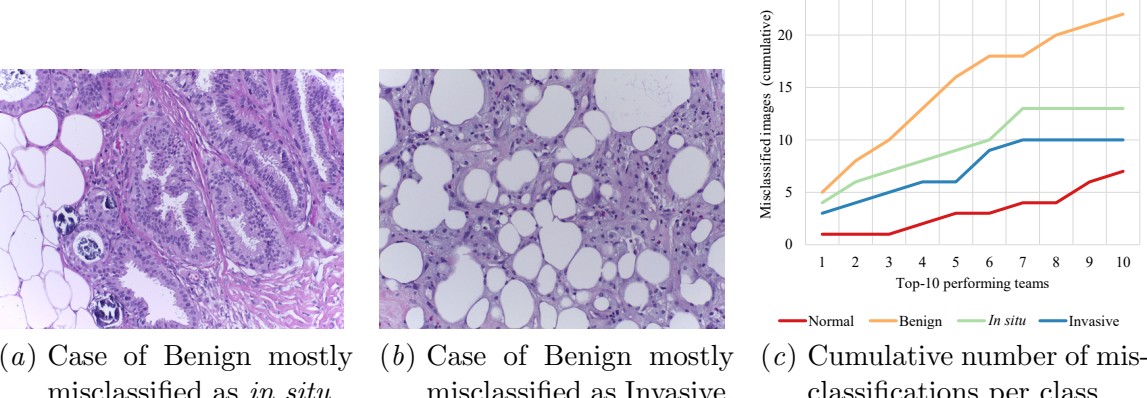

(a) Case of Benign mostly misclassified as *in situ*.

(b) Case of Benign mostly misclassified as Invasive.

(c) Cumulative number of misclassifications per class.

Figure 1: Examples of images misclassified by the top-10 methods of P-**A**.

## 2. Results

BACH received 64 effective submissions out of 677 registrations. The majority of the approaches were deep learning (DL)-based. Most methods followed a fine-tuning strategy of single or ensembles of DL networks, namely Inception (Szegedy et al., 2015), DenseNet (Huang et al., 2017), VGG (Simonyan and Zisserman, 2014) or ResNet (He et al., 2016).

P-**A** received 51 submissions. The top-10 accuracy was $\geq 0.8$, with a maximum of 0.87, whereas in their ICIAR submissions 8 out of the 10 top performing teams reported performances over 93%. The experts' average accuracy was $0.85 \pm 10$. The top methods used the entire image or large patches re-scaled to the expected network input size and without using staining normalization. The main differences between methods are related to the training scheme, namely data augmentation and model hyper-parameter adjustment. The most failed class was Benign, followed by *in situ*, Invasive and Normal (Fig. 1).

P-**B** had 13 participating teams. The top-3 Cohen's quadratic kappa score was $\geq 0.44$, with a maximum of 0.51. Due to the size of the images, the majority of the approaches did not opt for an out-of-the-shelf segmentation approach. Instead, the common strategy was to classify grid-sampled patches of the images and afterwards merge the predictions to obtain the pixel-wise classification. The best performing methods also opted by enriching the training data with images from P-**A**. Similarly to P-**A**, the approaches performed better for the Normal and Invasive, and worse for the *in situ* and Benign classes (Fig. 2).

## 3. Discussion and conclusions

Fine-tuning of deep networks was the preferred approach to solve BACH because it allows to achieve state-of-the-art performance while significantly reducing the required field knowledge (Litjens et al., 2017) and the number of training images (Tajbakhsh et al., 2016). On the other hand, unless the dataset is highly representative, fine-tuning from natural image models limits the performance of systems as the pre-trained features may not be specific enough for the task. Indeed, the Benign was the most challenging class for both parts, which is expected since the presence of normal elements and usual preservation of tissue ar-

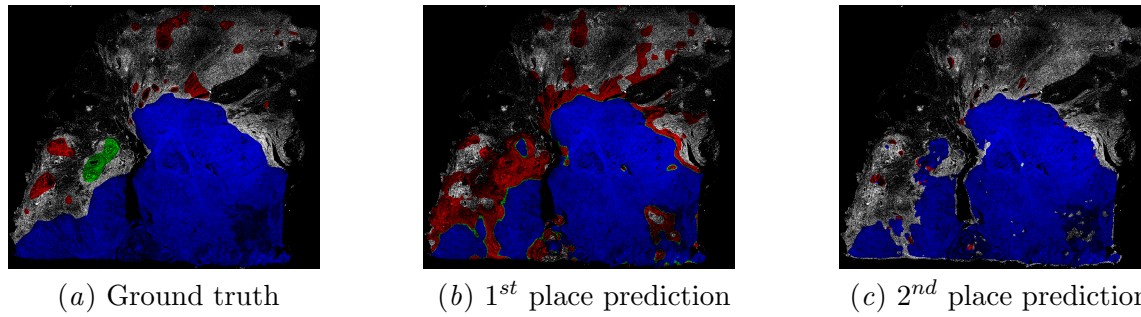

(a) Ground truth          (b) $1^{st}$ place prediction          (c) $2^{nd}$ place prediction

Figure 2: P-**B** test set predictions. ■ benign; ■ *in situ*; ■ invasive. The WSIs were converted to grayscale and the predictions overlayed (background was set to black).

chitecture associated with benign lesions makes this class specially hard to distinguish from normal tissue. Furthermore, the Benign class is the one that presents greater morphological variability and thus discriminant features are more difficult to learn.

For P-**A**, and unlike previous approaches for the analysis of breast histology cancer images (Araújo et al., 2017), the top teams used large patches or the entire image. This suggests that the overall nuclei and tissue organization are more relevant than nuclei-scale features for distinguishing different types of breast cancer. Interestingly, this matches the importance that clinical pathologists give to tissue architecture features during diagnosis. Also, DL approaches seem to be robust to small color variations of H&E images and thus color normalization may not be essential to attain high performance. The accuracy of the best methods is similar to the external specialists, showing that DL can achieve human-level performance for breast cancer biopsy microscopy classification. On the other hand, the performance discrepancy between the ICIAR report and the test-set shows a tendency of the participants to over-tune models to validation/test sets of known labels. Consequently, establishing rules for model validation in challenges (and publications) is essential to ensure that the reported results properly indicate the generalization capability of the models.

P-**B**'s image sizes made the task much more challenging, which lead to a reduction on the number of participants by inhibiting the direct application of pre-trained networks. This lead the participants to strive for more innovative solutions, namely on how to handle multiple scales and merge the predictions. Similarly to P-**A**, methods with higher receptive field tended to perform better, further enforcing the importance of tissue organization for the identification of pathological images.

Although the raw high performance of these methods is of interest, the scientific novelty of the approaches was overall reduced. Also, the black-box behavior of DL approaches hinders their application on the medical field, where specialists need to understand the reasoning behind the system's decision. It is the authors' belief that medical imaging challenges should further promote advances on the field by incentivating participants to propose significantly novel solutions that move from *what?* to *why?*. For instance, it would be of interest on future editions to ask participants to produce an automatic explanation of the method's decision. This will require the planning of new ground-truth and metrics that benefit systems that, by providing proper decision reasoning, are more capable of being used in the clinical practice.

## Acknowledgments

G.Aresta is funded by the FCT grant contract SFRH/BD/120435/2016. T.Araújo is funded by the FCT grant contract SFRH/BD/122365/2016. A.Campilho is with the project "NanoSTIMA: Macro-to-Nano Human Sensing: Towards Integrated Multimodal Health Monitoring and Analytics/NORTE-01-0145-FEDER-000016", financed by the North Portugal Regional Operational Programme (NORTE 2020), under the PORTUGAL 2020 Partnership Agreement, and through the European Regional Development Fund (ERDF).

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
