# OpenReview forum: "BACH: grand challenge on breast cancer histology images"
_MIDL.io/2019/Conference/Abstract — MIDL Abstract 2019_

### Official Review · AnonReviewer2 · 2019-04-29
**The uniqueness of BACH dataset needs to be highlighted more clearly.**

**Rating:** 2
**Confidence:** 2

**Review:**

The submission provides a summary of a recently held challenge on the authors' public dataset consisting of histology images of breast cancer tissues. The challenge gathered a reasonable amount of interests (51 submissions for the classification task and 13 for the segmentation task).

However, I have a couple of concerns.

1. It is not clear how the BACH dataset differs from existing similar datasets.
2. It is not clear what technical advancements lead to observed differences in performance across different models. Although this may be difficult, if you could highlight any key designs of the challenge devised to ensure controlled comparison, it would be great.

If point 1 could be addressed well, I would be happy to raise my score to 3.
Addressing 2 may be difficult given the space limit, however, would be of more interest to the MIDL audience.

---

### Official Review · AnonReviewer1 · 2019-05-01
**Summary of the BACH challenge**

**Rating:** 3
**Confidence:** 3

**Review:**

Interesting summary of the BACH challenge (also under review at MedIA), showing that fine tuned existing DL approaches performed well. Yet there are clear open problems with many benign regions being incorrectly labeled even for the top performing method.
Pretraining with natural images did not seem to help a lot, which is a surprising finding.
Less surprising is that medical applications (such as this one) necessitate more interpretative DL and the blackbox approaches employed in the BACH challenge.

---

### Decision · Program_Chairs · 2019-05-06
**Acceptance Decision**

Accept